# The Impact of Marine Engine Component Failures upon an Explosion in the Starting Air Manifold

**Leszek Chybowski** [1,*] , **Daniel Wiaterek** [2] **and Andrzej Jakubowski** [1]

1   Department of Machine Construction and Materials, Faculty of Marine Engineering, Maritime University of Szczecin, Ul. Willowa 2, 71-650 Szczecin, Poland
2   Centrum Innowacji Akademii Morskiej w Szczecinie, CIAM Sp. Z o.o., Ul. Starzyńskiego 9/102, 70-506 Szczecin, Poland
*   Correspondence: l.chybowski@pm.szczecin.pl; Tel.: +48-91-09-412

**Abstract:** Based on available sources, the frequency of explosions in the marine engine's starting air manifolds is determined under real conditions. A cause-and-effect analysis of these explosions and their root causes are identified. A probabilistic model of an explosion in the starting air manifold of a marine engine is built using a fault tree analysis (FTA). Using a stochastic simulation (Monte Carlo) and an exact reliability availability calculation (ERAC) algorithm applied to the developed FTA model, selected reliability measures are calculated to describe an incident of the top event, which involves an explosion in the starting air manifold. For such an event, several factors are calculated, including the availability, the unavailability, the failure frequency, and the mean time to failure. Based on the simulations, the relative frequency of the top event is determined in relation to the number of events that can simultaneously occur and lead to an explosion. The significance of each basic event is assessed to determine their individual impact on the explosion incident. The following measures are used: the Vesely–Fussell measure of importance, the criticality measure of importance, the Birnbaum measure of reliability importance, and the Birnbaum measure of structural importance. The results of the analysis show that defective starting air valves are most responsible for the explosion incident in the starting air manifold. During the first year of the ship's operation, the reliability does not fall below the value of 0.65, while the mean time to failure and the top event frequency are statistically at the level of one explosion per approximately 2.28 years of continuous engine operation.

**Keywords:** marine engine; starting air manifold; explosion prevention; fire safety; cause-and-effect analysis

## 1. Introduction

Due to the ever-increasing consumption of goods, there is an increasing demand for their transportation. Sea vessels are the most common means of transporting large volumes of goods [1,2]. As with any ship, their operation is associated with the possibility of damage and failure. According to reports from the Allianz Group, one of the main risks associated with ship operations is fires and explosions that result in accidents and incidents with an average frequency of 60 days [1]. Between 2009 and 2018, the listed organizations noted that 10% of the causes of all ship loss incidents were related to either a fire or an explosion. A large proportion of these cases are related to the operation of the engine room and their primarily internal combustion engines [3], which are estimated to account for 90–92% of the propulsion of modern ships in operation [4]; these are the primary source of energy for the ship's generator sets [5].

The use of modern high-powered internal combustion engines is inevitably associated with the risk of a fire or an explosion. The most important places such risks occur are during the operation of a low-speed crosshead engine; these are indicated in Figure 1. In particular, these relate to the crankcases, the surroundings of the fuel injection system, the exhaust

manifolds, the supply air accumulators, the sub-piston spaces, the turbochargers, and the starting air systems. The power rating of the largest, slow-speed, two-stroke crosshead engines exceeds 80 MW, which translates into the engine's dimensions and, potentially, the size of the consequences of its explosion [6]. Most marine engines—in fact, all large and high-powered engines—are starting to use compressed air. In addition, a significant part of the main propulsion engines is the reversible engines (which can operate in both directions of rotation for the drive shaft). In such a case, the switching between the directions of the rotation of the engine is performed by a compressed air system [7]. They are one of the engine sub-systems that can explode and lead to fires and joint explosions on the ship.

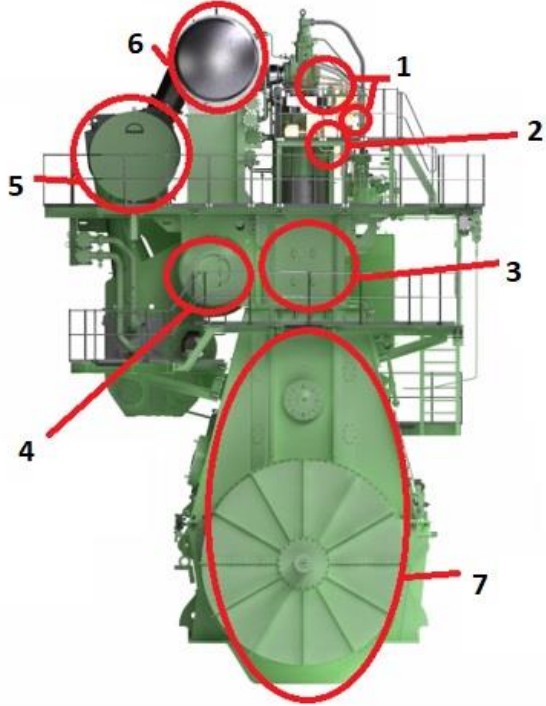

**Figure 1.** Components of a crosshead-type engine and its surroundings that can cause an explosion (modified based on Chybowski, 2022): 1—fuel injection system, 2—starting air system, 3—underpiston spaces, 4—scavenge space, 5—turbocharger, 6—exhaust manifold, 7—crankcase.

Historically, the first recorded case of an explosion due to an engine's starting air manifold that resulted in very serious consequences was an incident on the passenger liner RMMV Capetown Castle [8] (RMMV stands for Royal Mail Motor Vessel). The ship was equipped with two Harland and Wolff 10. 660/1500 DA main engines manufactured under license from B&W. These were 10-cylinder, two-stroke, slow-speed, reversible crosshead engines with a cylinder diameter of 660 mm and a piston stroke of 1500 mm. The ship was built in 1938, and the main engines listed were the largest engines built at that time. Each of them—with a height of 10 m (measured from the crankshaft axis), a length of 22 m, and a weight of about 910 tons—powered its own ship propeller [9].

The accident aboard the RMMV Capetown Castle occurred on 17 October 1960, during its entry maneuvers into the Las Palmas harbor. The explosion within the left main propulsion engine's starting air manifold led to the death of seven engine crew members: a senior engineer, a first engineer, a senior second engineer, a junior second engineer, two junior engineers, and a motorman. It later turned out that the causes of the accident were the poor condition of the starter valves and the lack of established procedures for clearing debris in the piping systems. The ship used pressurized hydraulic oil for this purpose, which involved using a portable pump to feed the hydraulic presses; such a pump is normally used to apply tension to the engine bolts during maintenance work that requires the removal and installation of engine components.

The accident drew the attention of representatives of companies and organizations involved in maritime transportation to the need to increase safety and to minimize the risk of an explosion of an engine's starting air system components and, additionally, to minimize the consequences of a potential explosion. The requirements for the explosion-proofing of starting systems are found in the SOLAS International Convention for the Safety of Life at Sea (1/II-1/C/Regulation 34) [10]. The incident also resulted in the development, and later, in 1972, implementation, of the International Association of Classification Societies (IACS) guidelines in this area, covered in sections M11 and M59/M59.6/6.1 [11], which were subsequently implemented by classification societies [11].

According to the aforementioned regulations, the compressed starting air pipelines on an internal combustion engine must be equipped with a check valve. For compression-ignition internal combustion engines with a cylinder diameter of not less than 230 mm, it is required that the starting system should be equipped with a bursting plate (bursting disc) or a flame-arresting device (flame catcher). For reversible marine engines with a main starting manifold, the mentioned regulations require such protection at each stub, which supplies compressed air into the starting valves, and, for non-reversible engines, at the inlet to the starting air manifold. In addition, for dual-fuel engines fueled with gaseous fuel, the air intake piping to each cylinder should be equipped with effective flame arrestors. Moreover, to increase safety, additional measurements and monitoring systems can be applied (see: Patents Section).

### 1.1. Background and Aims of the Research

A detailed presentation on the explosion statistics for the starting air manifolds has not yet received a comprehensive study in the form of a report or a scientific publication. This fact can be attributed to the relatively infrequent occurrence of these explosions, the dispersion of information, and the non-publication of information based on the events. Nevertheless, a small number of available literature sources show that these explosions occur on ships of different types, which carry different cargoes and operate in various waters. The number of described accidents that involve humans have decreased, but explosions of the engine's starting systems still occur. For example, in a publication by Song et al. [12], the authors report accidents that occurred in the years prior to 2000 (the date of their publication) that began with an air manifold explosion. This article included a case involving an explosion in the main propulsion engine of a VLCC-type tanker in Tokyo Bay (VLCC stands for Very Large Crude Carrier), which resulted in the ship losing its propulsion. Luckily, there were no collisions or leaking of the oil that formed part of its cargo.

The aforementioned authors also pointed to other cases recorded by the classification societies in which explosions occurred in the starting air manifolds [12], including reports between 1989 and 1998 from the British classification society Lloyd's Register of Shipping (LR), which recorded 11 cases of explosions of this type [12]. Of course, it should be borne in mind that, for incidents in which people have not been hurt, dangerous events are often not reported to the classification society by the shipowner (these are the so-called unreported cases).

When analyzing accidents and disasters, post-accident reports and service recommendations from engine manufacturers are valuable sources of information. However, they often do not have complete information on the exact date of the incident, the ship's name, and the engine type. However, they do provide some insight into the regularity of the patterns of the behavior, the root causes, and the potential consequences of such explosions. Figure 2 shows the annual distribution for the number of explosions in the starting air manifolds, from the middle of the last century until now, for the events that the present authors found [12–19]).

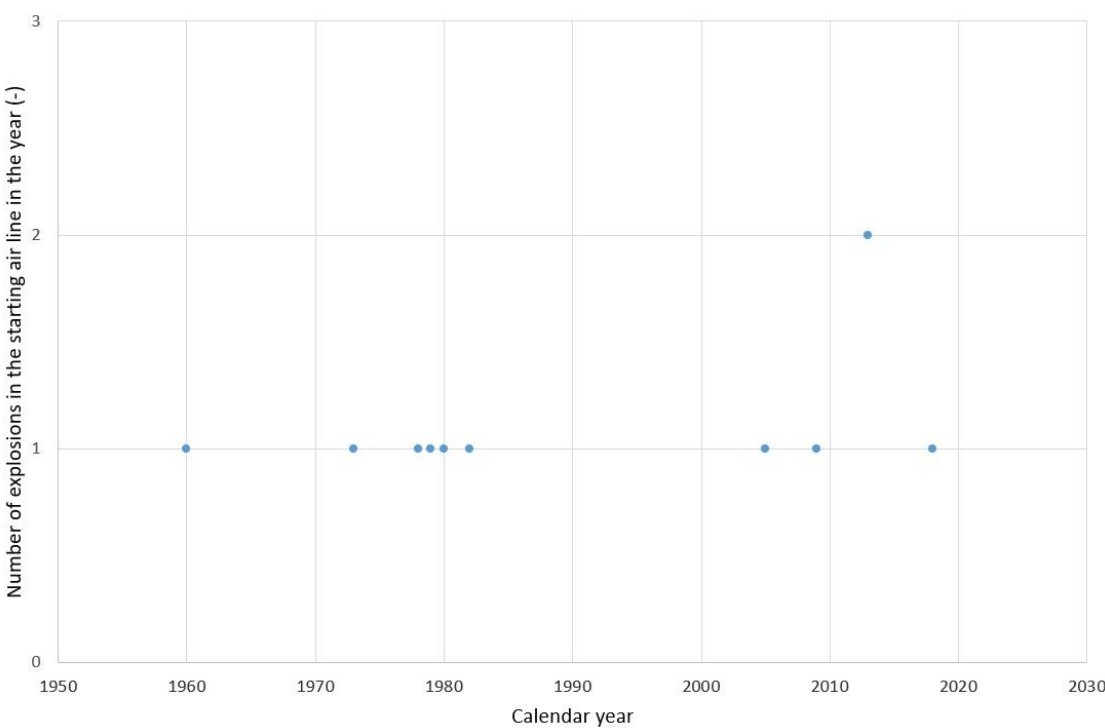

**Figure 2.** Annual distribution for the number of explosions in the starting air manifold, determined from events extracted from the literature search on the topic (own research).

Appendix A summarizes the primary information (to the extent that it is available in the source materials) about the incidents of explosions in the starting air manifolds, which are included in Figure 2. Appendix A provides the exact or approximate date of the incident, information about the ship, engine, causes, effects, and a literature source that can deepen our knowledge of the incident.

The cases of explosions of a marine engine's starting air manifolds, analyzed by the present authors, can be a source of extreme danger in close vicinity to the engine. From an analysis of the statistics and the listed and available reports of maritime disasters on the Internet [1,15,17,20–22], it is clear that the topic is worthy of a detailed cause-and-effect analysis. Given the relatively small number of scientific articles devoted to the root causes of explosions in the starting air manifolds, the authors undertook to prepare such an analysis, which is presented in this article. The literature analysis was prepared and a probabilistic model which takes into account causes and effects related to starting line explosions was built. The model was used to analyze root causes and build an importance ranking of basic events that can lead to explosions.

### 1.2. Causes of an Explosion in the Starting Air Manifold

Explosions are caused by rapid, exothermic chemical reactions (e.g., combustion), especially branched chain reactions, and violent physical phenomena (e.g., explosions of steam boilers and gas cylinders) [23]. In the case of explosions in the starting air manifolds, we deal with combustion phenomena. As a result of the explosion in the starting air manifold, the components of the engine's starting system, which are exposed to the dynamic load of internal forces associated with the sudden increase in pressure during the explosion process, may burst. Combustion is a redox chemical reaction that combines a combustible body (fuel) with an oxidizer, which is accompanied by the release of heat and light. The combustion reaction is triggered by an initiating agent that vaporizes the fuel and creates intermediate chemicals [24].

A redox reaction, or oxidation–reduction reaction, is a chemical reaction in which electron transfer occurs between two compounds; the latter is referred to as an oxidant

and a reductant. This process results in a change in the degree of oxidation of the elements within these compounds. An oxidant is an element, or chemical compound, that accepts an electron in the redox reaction under analysis. Oxidants in reactions decrease their degree of oxidation (i.e., undergo reduction) and simultaneously cause an increase in the degree of oxidation of the remaining products of the reaction. The oxidant in the combustion process can be pure oxygen, oxygen from the air, substances containing sufficient oxygen within their molecule, or other compounds or elements with oxidizing properties, such as chlorine or bromine [24]. Combustible materials are substances that, when heated, emit gases in sufficient quantity to permanently ignite them [25]. Oxygen, on the other hand, is one of the most active chemical elements that react with many elements and chemical compounds. If this process occurs rapidly, then it can be accompanied by light effects and high temperatures [23].

The contemporary combustion model—in addition to fuel, oxidant, and energy source—involves free radicals, i.e., molecules with unpaired electrons formed by chemical reactions, which promote the formation of a branched chain reaction for the combustion [26]. Nevertheless, to simplify it as much as possible, for the model built for the probabilistic root cause analysis of explosions in supply air manifolds, the impact of free radicals is not mapped as separate basic events. In the case of an explosion in the starting air manifold, the combustible material inside the manifold and associated components may be [27]:

- fuel or cylinder oil that has entered from a particular cylinder system through a leaking or blocked-in-open starting valve while the engine is running, or during engine starting, if the fuel (from a defective injector) or the cylinder oil (from a defective or improperly used automatic cylinder liner lubrication system) has accumulated in the cylinder during idling;
- fuel or oil that has collected on the head as a result of leaks (routinely check for leaks in the fuel and oil system) and has entered the starting system through an improperly sealed starting valve as a result of using non-original O-rings;
- lubricating oil fed from the starting air compressor, which results from a malfunctioning oil separator and worn compressor piston scraper rings, and as a consequence of the compressor sucking in air that contains oil vapor;
- lubricating oil from the components of the starting system (handled not in accordance with the manufacturer's instructions, which outlines the strict requirements for the lubrication/non-lubrication of the equipment components in the starting system) that was mixed with the starting air during the engine start-up;
- carbon deposits formed from the thermal decomposition of the above-mentioned petroleum substances on the heated walls of the starting air manifold and its associated components.

The energy source required to initiate the ignition can be external or related to the thermodynamic state of the combustible mixture (a self-ignition). In the case of an external agent, the initiator of an explosion in the marine engine's starting systems can involve hot gases with a temperature greater than 1200 °C (for a combustible mixture from the cylinder and exhaust gases). These gases can enter the starting valve supply line and the engine's starting air manifold through the starting valve. This situation may occur when one of the starting valves is leaking. Then there is an outflow of hot gases into the starting air manifold. A view of the starting valve, which is removed after the supply air manifold has exploded, is shown in Figure 3 [16].

The possibility of the initiation of an explosion due to the spontaneous ignition of a combustible mixture was presented by the classification society ClassNK (Nippon Kaiji Kyokai) [27], whose staff forwarded the hypothesis that accumulated oil or fuel in the starting manifold could spontaneously ignite during a start-up of the engine [28]. According to this hypothesis, the mixture of combustible substance and starting air is ignited, and a high temperature arises due to the sudden additional compression of air flowing into the starting air manifold. The pressure in the starting air tanks is usually 2.5–4.2 MPa. The airflow into the starting system during the engine start-up may temporarily significantly increase its

pressure (for which a pressure peak is observed). The latter is associated with a growth in the air temperature to around 400 °C, which has been confirmed experimentally [28] and is a sufficient temperature to initiate an explosion.

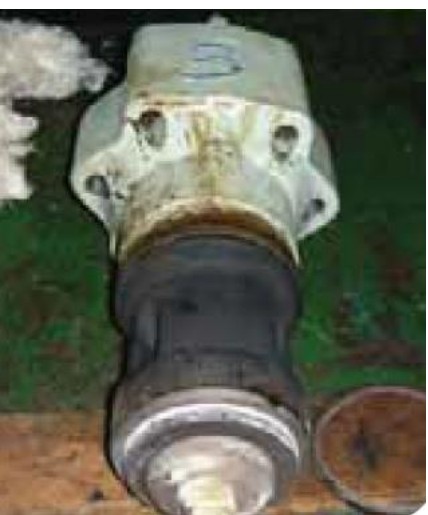

**Figure 3.** Photograph of the starting valve removed from the engine after an explosion in the starting air manifold (Royal Belgian Institute of Marine Engineers 2009).

### 1.3. Effects of an Explosion in the Starting Air Manifold

In a situation where the engine does not have the safety components required by law or where they are unfit (due to improper maintenance, negligence, or the use of uncertified replacement parts), an explosion may result in the bursting of the starting air manifold and its associated equipment. Examples of the effects of explosions in the starting air manifolds are shown in Figure 4.

As a result of the explosion, the following mechanical parts were damaged:

- starting air manifold, including the bursting of the tube and/or breaking of the blanking cover;
- air supply pipes to the engine cylinder starting valves, which can be completely destroyed or torn locally;
- starting valves, including the valve bursts;
- other objects close to the engine's starting system (i.e., the interior of the engine cylinder, the engine hull, the fuel lines, the platform plates, etc.).

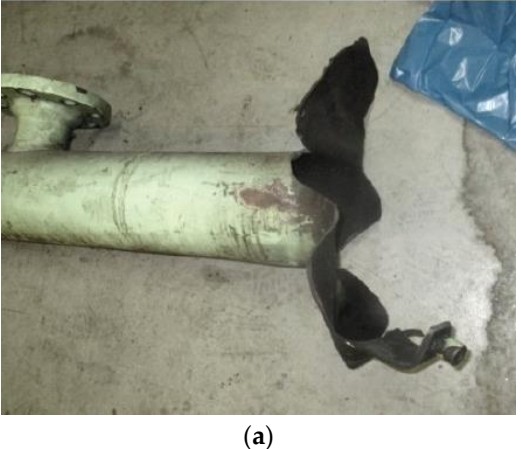
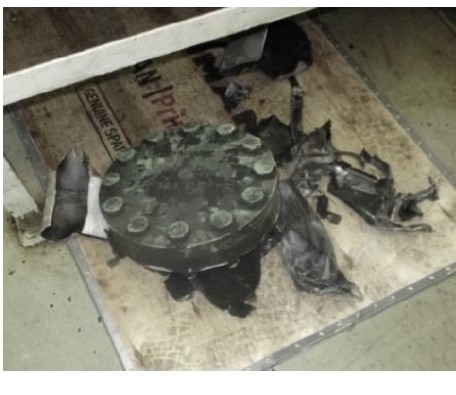

(**a**)　　　　　　　　　　　　　　　　　　　　　　　　　(**b**)

**Figure 4.** *Cont.*

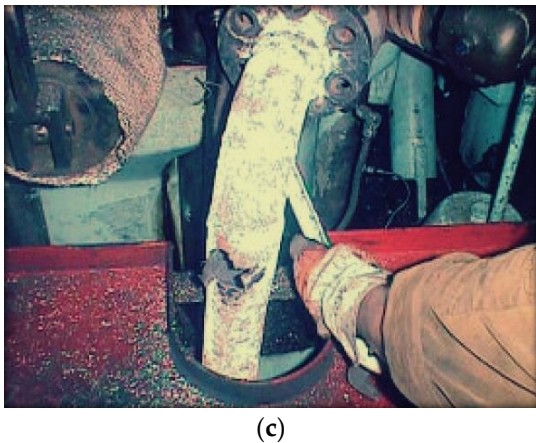
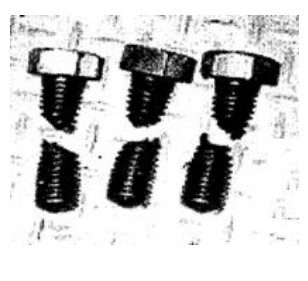

(**c**)                                                    (**d**)

**Figure 4.** Examples of the effects of an explosion in a starting air manifold: (**a**) torn manifold main line(MAN, 2018); (**b**) torn manifold flap (MAN, 2018); (**c**) torn air supply line to the starting valve (Officer of the Watch, 2013); (**d**) bolts holding the starting valve head damaged by the explosion (modified based on Song & Sasaki, 2003).

Metal parts inside the starting air manifold that are detached by the explosion may be scattered with a great force near the explosion site, posing a danger to the life and health of the machine crew members. Especially dangerous are potential injuries to the internal organs and the head [29]. Engine components that are detached as a result of the explosion can also cause further (secondary) mechanical damage to the interior of the engine room (damage to the machinery and equipment, the secondary fires in the engine room, etc.) [30]. In addition, torn starter valves, the fragments of which remain inside the cylinder, can cause secondary damage to the engine (i.e., the cylinder head, the piston, the cylinder liners, the fuel injectors, etc.).

## 2. Materials and Methods

There are many tools used for safety and reliability analysis, including simulation methods [29–32]. We used fault trees, for which we performed qualitative and quantitative analyses. To conduct a probabilistic analysis of the factors that influence the occurrence of an explosion in the starting air manifold, a model was built in the form of the fault tree; this is shown in Figure 5. A fault tree analysis is based on a deductive decomposition of events occurring during the system operation [33–35]. Any change in the structure and functioning of the components of the system and its environment is treated as an event [36]. The fault tree start-up is used to map the logical combination of basic events (not subject to further decomposition at the adopted level of detail of the analysis conducted) that leads to the occurrence of subsequent intermediate events and, finally, to the top event. Targeted by this analysis, the top event is an explosion in the starting air manifold.

It is assumed that the object is modeled as a slow-speed, high-powered crosshead marine engine [37,38]. The reliability models were also adopted for such an engine, characterizing the individual basic events in the fault tree. The CARA FaultTree Application v. 4.1. Academic Edition (Sydvest Software, Trondheim, Norway) supported the calculations. To increase the readability, the model was divided into two subtrees that are connected by a transfer (T1).

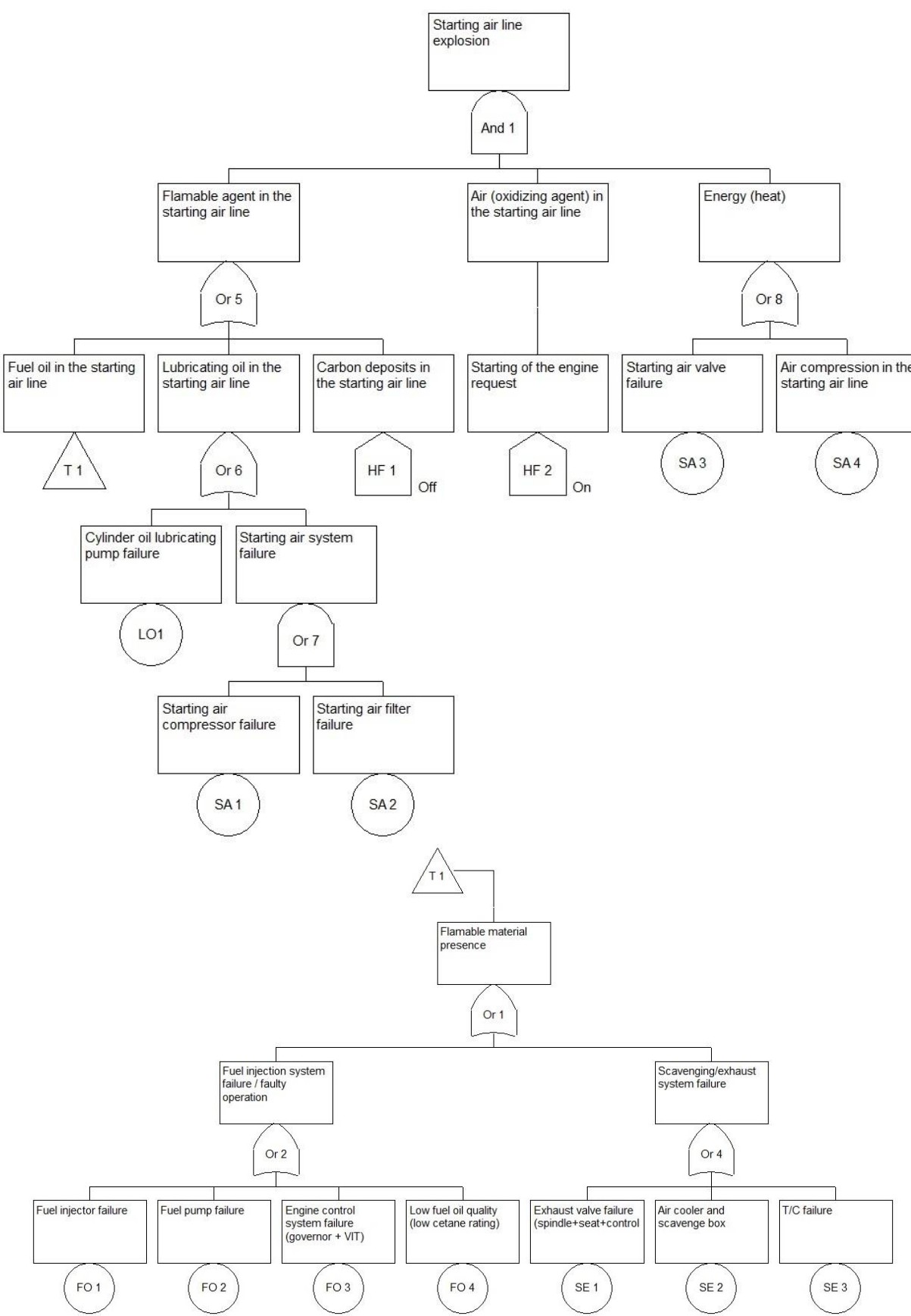

**Figure 5.** Fault tree modeling of the explosion in the starting air manifold of a slow-speed, internal combustion crosshead engine.

The tree was based on the initial assumptions that are presented in the introduction, i.e., referring to the basic factors of the so-called fire triangle: a combustible substance, an oxidizer, and a heat source. Each of these factors was sequentially decomposed in terms of sub-systems and engine components, whose malfunction determines the occurrence of each factor. In particular, the model captures the starting air system, the fuel supply system, the cylinder liner lubrication system, the charge exchange system, and the components responsible for the engine control. Factors such as the presence of carbon deposits inside the starting air manifold, the occurrence of instantaneous air compression phenomena inside the starting air manifold (hypothesized by ClassNK), and a large fuel auto-ignition delay were included in the model. It was also assumed that the presence of carbon deposits inside the starting air manifold is negligible due to various routine inspections. The reverse situation can be analyzed in further studies by changing the status of the event modeling from OFF to ON. In addition, the tree does not include safety devices, such as bursting discs and safety valves, since the intended purpose of the developed model was to represent the occurrence of an explosion. These listed devices do not affect the occurrence of an explosion since they only reduce the negative effects of an explosion once it occurs.

Basic events in the fault tree were modeled using the event types and the quantitative description shown in Table 1. Information on the reliability of the individual functional components and engine sub-systems was taken from the literature [37,38], while mean times to repair were estimated by considering the time inputs indicated in the engine manuals [39–41]. Simultaneously, it was assumed that all the necessary parts for the repair were available on board the ship. Most sub-systems were modeled on the fault tree as repairable objects with an exponential distribution of time to failure with damage intensity $\lambda$ and a mean time to repair $T_r$. In addition, the model includes two external (two-state) events, one event modeled by the frequency of occurrence $f$ and one on-demand event described by a fixed probability of occurrence (a fixed value of unavailability).

The developed probabilistic model was used to calculate the reliability measures characterizing the top event (the explosion in the engine's starting air manifold) and to determine the impact of individual basic events on the main incident (the component importance analysis). The measures that describe the top event were defined as follows (definitions for the top event/system and basic events/system components are analogous):

- System unavailability $Q_0(t)$—the probability that the top event occurs at time $t$. In the case analyzed here, the unavailability determined for the top event establishes the probability that at time $t$ an explosion in the starting air manifold will not occur with the assured operation of the system components, according to the event description of the repairable components.

- System availability $A_0(t)$—the probability that the top event is not occurring at time $t$. This measure is associated with the value of $Q_0(t)$, i.e., via $A_0(t) + Q_0(t) = 1$.

- System reliability $R_0(t)$—the probability that the top event has not occurred in the time period from 0 to $t$. Concerning the repairable objects, reliability describes the situation of using the system until the first failure. In other words, in the case analyzed here, the reliability determined for the top event ascertains the probability that, in the period [0,t], an explosion in the starting air manifold will not occur in the absence of maintenance of system components, i.e., the probability that the system has survived up to time $t$.

- Mean time to failure of system $T$—the mean time to the first occurrence of the top event. In the present case, it is the average time measured from the start of engine operation until the explosion occurs in the starting air manifold.

- Frequency of the top event $f_0$—this is used to describe events that occur now and then but with no duration. Thus, the probability that the event occurs at time $t$ equals zero.

**Table 1.** Characteristics of the basic events involved in the occurrence of an explosion in the starting air manifold of a slow-speed, internal combustion crosshead engine.

| Event Name | Event Type | Parameter | Value | Description |
|---|---|---|---|---|
| FO 1 | Repairable | $\lambda$ (1/($10^6$ h))<br>$T_r$ (h) | 0.000041<br>3 | Fuel injector failure |
| FO 2 | Repairable | $\lambda$ (1/($10^6$ h))<br>$T_r$ (h) | 0.000035<br>3 | Fuel pump failure |
| FO 3 | Repairable | $\lambda$ (1/($10^6$ h))<br>$T_r$ (h) | 0.00002<br>12 | Engine control system failure (governor + VIT) |
| FO 4 | On-demand | $q$ | 0.005 | Low fuel oil quality (low cetane rating)<br>Carbon deposits in the starting airline |
| HF 1 | House | External event | Off | Assumed that the standard maintenance is performed (including periodic cleaning of the manifold and checking the functionality and proper use of the starting air manifold drainage system) |
| HF 2 | House | External event | On | Start of the engine request<br>Assumed that the start of the engine is requested |
| LO1 | Repairable | $\lambda$ (1/($10^6$ h))<br>$T_r$ (h) | 0.000033<br>24 | Cylinder oil lubricating pump failure |
| SA 1 | Repairable | $\lambda$ (1/($10^6$ h))<br>$T_r$ (h) | 0.0001<br>24 | Starting air compressor failure |
| SA 2 | Repairable | $\lambda$ (1/($10^6$ h))<br>$T_r$ (h) | 0.000066<br>6 | Starting air filter failure |
| SA 3 | Repairable | $\lambda$ (1/($10^6$ h))<br><br>$T_r$ (h) | 0.00001<br><br>2 | Starting air valve failure (including blockage of the valve in the open position, large blowouts from the combustion chamber to the starting air manifold with flame arresters installed, and small blowouts in the absence of flame arresters) |
| SA 4 | Frequency | $f$ (1/$10^6$ h) | 0.0041666 | Air compression in the starting airline (it was assumed that there are ten consecutive starts during each ship maneuvering, and the maneuvers are carried out daily (1 × daily)) |
| SE 1 | Repairable | $\lambda$ (1/($10^6$ h))<br>$T_r$ (h) | 0.000027<br>12 | Exhaust valve failure (spindle + seat + control) |
| SE 2 | Repairable | $\lambda$ (1/($10^6$ h))<br>$T_r$ (h) | 0.000004<br>12 | Air cooler and scavenge box |
| SE 3 | Repairable | $\lambda$ (1/($10^6$ h))<br>$T_r$ (h) | 0.000005<br>48 | T/C (turbocharger) failure |

The exact reliability/availability calculation (ERAC) algorithm developed by Terje Aven [42] and the stochastic Monte Carlo simulation, created by Nicholas Metropolis and Stanislaw Ulam [43], were used to determine the mentioned ratios.

Assuming a fault tree with *n* independent input events, let $\vec{y} = (y_1, y_2, \ldots, y_n)$ denote the random state vector of the input events, where $y_i$ is equal to 1 when the *i*-th input event occurs (otherwise, it is 0). If *A* denotes a set of all the states *y* of the fault tree, such that the top event occurs, the unavailability system is given by [42]:

$$Q_0 = \sum_{y \in A} \Pr[(t) = \vec{y}] \tag{1}$$

After substituting $\Pr[y_i(t) = 1] = q_i(t)$ and $\Pr[y_i(t) = 0] = 1 - q_i(t) = p_i(t)$, the formula on which the ERAC algorithm is based is obtained:

$$Q_0 = \sum_{y \in A} \left\{ \prod_{i=1}^{n} [p_i(t)^{1-y_i} q_i(t)^{y_i}] \right\} \tag{2}$$

On the other hand, the Monte Carlo procedure is one of the methods used to solve mathematical problems based on fitting a random process to the problem that is to be solved, whose statistical parameters would approximate the sought solution values. In the present case, stochastic simulation estimates the system's reliability and other reliability indicators. The output parameters and the data for the simulations are summarized in Table 2.

**Table 2.** Parameters of the performed simulations.

| Parameter | Value |
|---|---|
| Top event | AND 1 |
| Maximum cut set size (-) | 4 |
| Modularization level (-) | 0 |
| Mission time (h) | 8760 |
| Number of simulations (-) | 2000 |
| Seed for simulation (-) | 5355 |

The analysis was carried out in relation to the top event marked as the output from AND 1 gate in the model shown in Figure 5. The simulation was run for the assumed one year of the system operation. We have carried out the maximum number of simulations provided by the computer software and maximal size of cut sets available in the analyzed fault tree.

The main parameters obtained from the simulation execution are the simulation time interval $[0,t)$ and the number of simulation runs. In each run, a realization of the system's performance at time intervals $[0,t)$ is simulated, and the moments for which the top event occurs are recorded. The value of the reliability function is estimated as the relative number of launches in which no top event occurred during the time interval $[0,t)$. In turn, the sum of the times to the first system failure for the start-up, when the top event occurred, divided by the number of simulations (in which the top event occurred), determines the mean time to system failure *T*. Moreover, the number of simulations in which the top event occurred per mission time set for the simulation is used to determine the frequency of the top event and the frequency distribution of the top event due to the number of simultaneous basic events.

To assess the impact of the influence of the identified factors on the occurrence of an explosion in the starting air manifold, the measures of importance were determined, which were used to discover the most significant events that led to the undesirable situation (i.e., the explosion). The following measures of importance for the basic events were calculated [44–46]:

- Vesely–Fussell's measure of importance $I^{V\text{-}F}$, which can be interpreted as the probability that the top event is caused by the *i*-th input event (at time *t* with probability $\check{Q}_j(t)$ the *j*-th cut set of the tree containing the *i*-th basic event occurs), when it is clear that the top event has occurred, that is:

$$I_i^{VF}(t) \approx \frac{\sum_{j=1}^{m_i} \check{Q}_j(t)}{Q_o(t)} \qquad (3)$$

- Criticality (Lambert's) measure of importance $I^C$, which can be interpreted as the probability that the *i-th* component is critical for the system (occurring at time *t* with probability $q_i$ failure of the component causes system failure) and failed at time *t*, given that the system failed at time *t*, which is described by the formula:

$$I_i^C(t) = \frac{I_i^B(t) \cdot q_i(t)}{Q_0(t)} \qquad (4)$$

- Birnbaum's measure of reliability importance $I^B$ can be interpreted as the difference between the probabilities of the top event computed under the assumptions that the input event number *i* is known to occur and is known not to occur, respectively, that is:

$$I_i^B(t) = Q_o[t|q_i(t) = 1] - Q_o[t|q_i(t) = 0] \tag{5}$$

- Birnbaum's measure of structural importance $I^{Bs}$ for component $i$ is defined as the relative number of system states for which component $i$ is critical for the system. The structural Birnbaum measure is equal to the reliability Birnbaum measure calculated for the situation in which the unavailability of all components is $q_i(t) = 0.5$, which can be represented in the form:

$$I_i^{Bs}(t) = I_i^B[t|q_1(t) = q_2(t) = \ldots q_n(t) = 0.5] \tag{6}$$

The listed importance measures were calculated, based on the unavailability estimated, using the ERAC algorithm that was mentioned earlier.

### 3. Results and Discussion

Using the model built and described earlier (in the form of a fault tree) and the characteristics of each basic event, simulations were performed to qualitatively and quantitatively assess the factors that lead to an explosion. As a result of the qualitative analysis, minimum cut sets of the analyzed tree were identified (a cut set is a set of events that, while occurring at the same time, cause the top event to occur; a cut set is minimal if it cannot be reduced by any of the components without losing the status of the cut set), including for the following analyzed tree occurrences:

- 16 cut sets with two components: {SA 3,FO 1}, {SA 4,FO 1}, {SA 3,FO 2}, {SA 4,FO 2}, {SA 3,FO 3}, {SA 4,FO 3}, {SA 3,FO 4}, {SA 4,FO 4}, {SA 3,SE 1}, {SA 4,SE 1}, {SA 3,SE 2}, {SA 4,SE 2}, {SA 3,SE 3}, {SA 4,SE 3}, {SA 4,SE 3}, {LO1,SA 3}, {LO1,SA 4}.
- Two cut sets with three components: {SA 1, SA 2, SA 3}, {SA 1, SA 2, SA 4}.

The basic quantitative indicators and estimated standard errors, determined for the top event of the analyzed tree, are provided in Table 3.

**Table 3.** Reliability measures calculated for the top event of the analyzed tree.

| Parameter | Calculation Based on the ERAC Algorithm | Calculation Based on the Monte Carlo Simulation | Standard Error |
|---|---|---|---|
| Unavailability $Q_0(t)$ (-) | $1.37440 \times 10^7$ | 0 | $1 \times 10^{-7}$ |
| Availability $A_0(t)$ (-) | $9.99999 \times 10^{-1}$ | 1 | $1 \times 10^{-7}$ |
| Mean number of failures within $t = 8760$ h | – | 0.639 | 0.0300548 |
| Mean time to failure $T$ (h) | – | 20852.4 | 797.897 |
| Failure frequency $f_0$ (failure/h) | – | $7.29452 \times 10^{-5}$ | $3.43091 \times 10^{-6}$ |
| Reliability $R_0(t = 8760$ h) | – | 0.65850 | 0.00001 |

The average system availability, which is determined by both methods, is close to or equal to the value of 1 (while the system unavailability is close to or equal to zero). This factor is due to the relatively low probability of the individual basic events, with a short mean time to repair that does not exceed 48 h. The change in the system reliability for the first year of engine use is shown in Figure 6. The employment of reliability as a measure to describe the explosions, and, more specifically, the assumption of the lack of implementation of maintenance work (i.e., treating the system as unrepairable), makes it possible to assess the length of time until the first failure. During the first year of the ship's operation, the reliability does not fall below the value of 0.65, while the mean time to failure and the top event frequency indicated in Table 1 are statistically at the level of one explosion per approximately 2.28 years of continuous engine operation. This result confirms the reality of the threat that is the subject of this article.

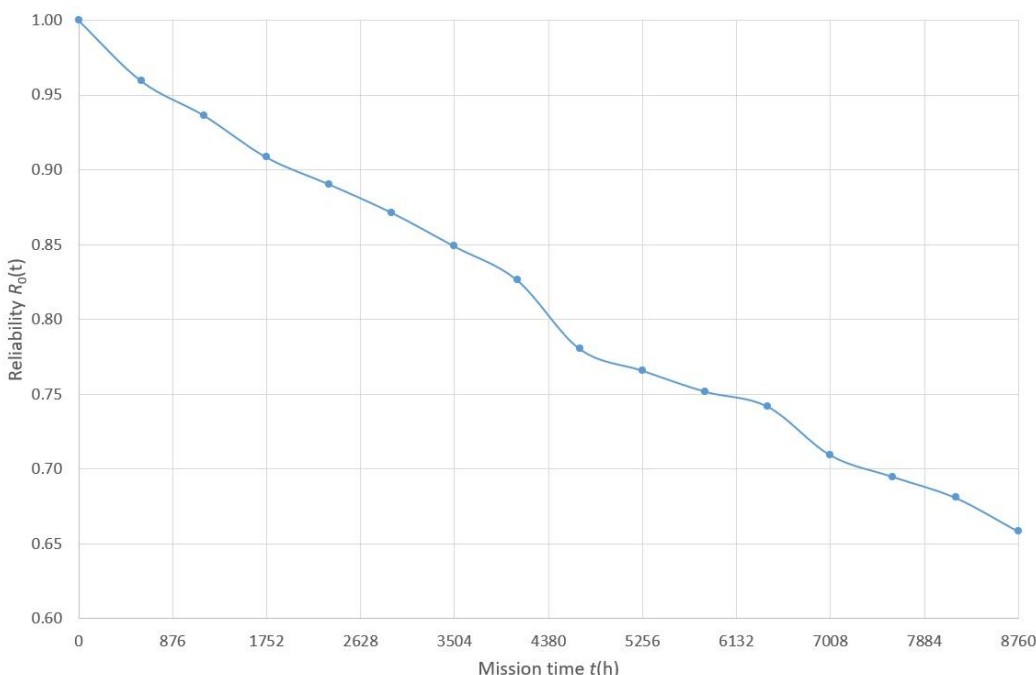

**Figure 6.** Probability of an explosion in the starting air manifold during the first year of engine operation (to the first failure).

A result of the simulations is a determination of the frequency distribution for the top event due to the number of basic events initiated during a given simulation, as shown in Figure 7. The results show that 65.75% of the simulations did not result in the occurrence of the top event, while for the simulations in which the top event occurred in 19.50%, 10.15%, and 4.60% of the simulations, the top event occurred once, twice, and more than twice during the analyzed mission time, respectively.

Analyzing the reliability, the mean time to failure, and the failure frequency values obtained from the stochastic simulations, it can be concluded that the apparent rarer occurrence of the explosions observed in real conditions (for which no media reports, or reports of any kind, are available) compared to the simulation can be explained as follows:

- The assumptions made in the model, and the focus on the occurrence of the top event, in qualitative terms (for which the corresponding severity of the explosion drivers could be described in more detail, which unfortunately requires additional information). This means that the occurrence of necessary factors may not always result in the worst possible scenario.
- A relatively high number of engine starts is assumed in the simulation, which approaches the worst-case scenario (the engine was assumed to be started 3650 times per year).
- The lack of complete information on the actual incidence of explosions in the starting air manifold (with properly functioning safeguards, an explosion in the starting air manifold that results in the bursting of a disc, or the activation of a safety valve, will almost always not be reported anywhere as a natural part of the operational process).
- Ongoing inspection of the technical condition of the machinery and equipment by the ship's crew ensures a high level of operational efficiency and safety, including the avoidance of dangerous situations associated with negligence in the implementation of planned maintenance work and also acting in accordance with the most current recommendations of the engine manufacturer (adherence to the indications contained in the engine manufacturer's service letters).

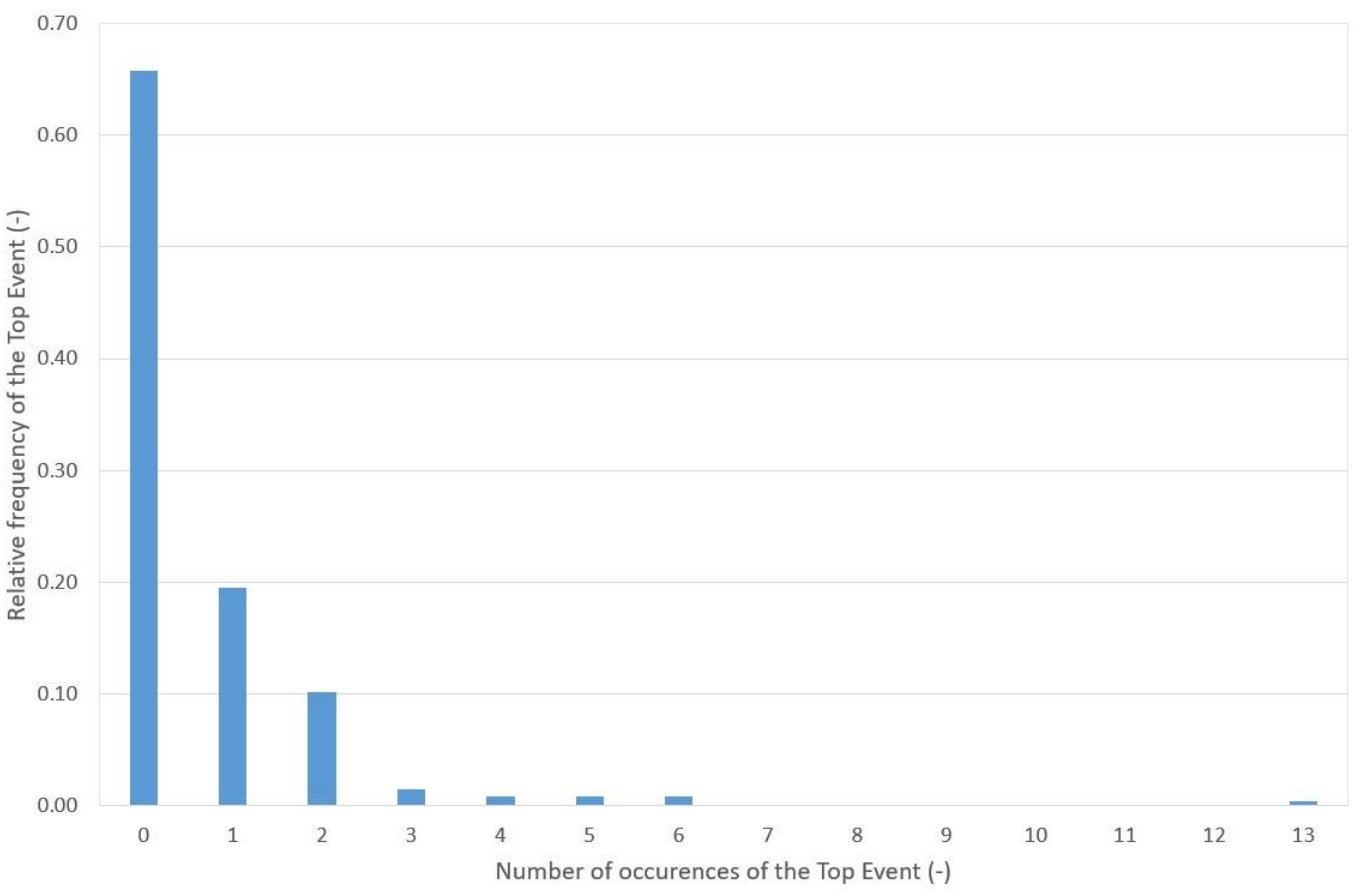

**Figure 7.** Relative frequency of the top event.

The final element of the conducted analysis was to determine the importance of individual events and their impact on the occurrence of the top event. The results of the analysis are shown in Figure 8. The individual events were ranked (from the left-hand side of the graph) from the most significant to the events with the least impact on the probability of an explosion in the starting air manifold.

The analysis shows that, considering all the importance measures used, the most significant events are SA3—a starting air valve failure—and FO4, poor fuel self-ignition properties (a low cetane rating). These results from the damage analysis reflect the root causes found in real cases of explosions in starting air manifolds (see Appendix A). This fact confirms the reliability of the model and the consistency of the simulation results obtained with those obtained from post-accident testing of damaged engines under operating conditions.

Attention should be paid to event SA4 (air compression in the starting airline), which in the case of the Birnbaum measure of the structural $I^{Bs}$ is located in the second position that closely follows SA3. Nevertheless, the high value of this measure is due to the location of this event in the structure modeled by the unfitness tree. Due to the way the SA4 event was modeled as a frequency event, $I^{V-F}$ and $I^C$ values could not be determined for this event. In contrast, considering the individual probabilities of the occurrence of individual events, other than the analyzed event (Birnbaum reliability measure, $I^B$), it is indicated that this event does not stand out significantly from all the others ($I^B$ for event SA4 has a value close to the others, among the events located on the left-hand side of the graph).

The next positions on the chart are occupied by events with a lower impact than SA3 and FO4 for the occurrence of the explosion (around a 7–10 times lower probability of impact for the occurrence of the top event), i.e.,:

- LO1 (cylinder oil lubricating pump failure);

- with similar SE1 notes (exhaust valve failure), FO3 (engine control system failure), and SE3 (T/C failure);
- with similar FO1 notes (fuel injector failure) and FO2 (fuel pump failure).

Events SE2 (inlet air cooler and scavenge box), SA2 (starting air filter failure), and SA1 (starting air compressor failure) have the least impact on the occurrence of the top event. The low impact of the listed events is due to their complex contributions to the occurrence of the hazardous situation, which relates to the necessity of several basic events with relatively low probabilities of occurrence.

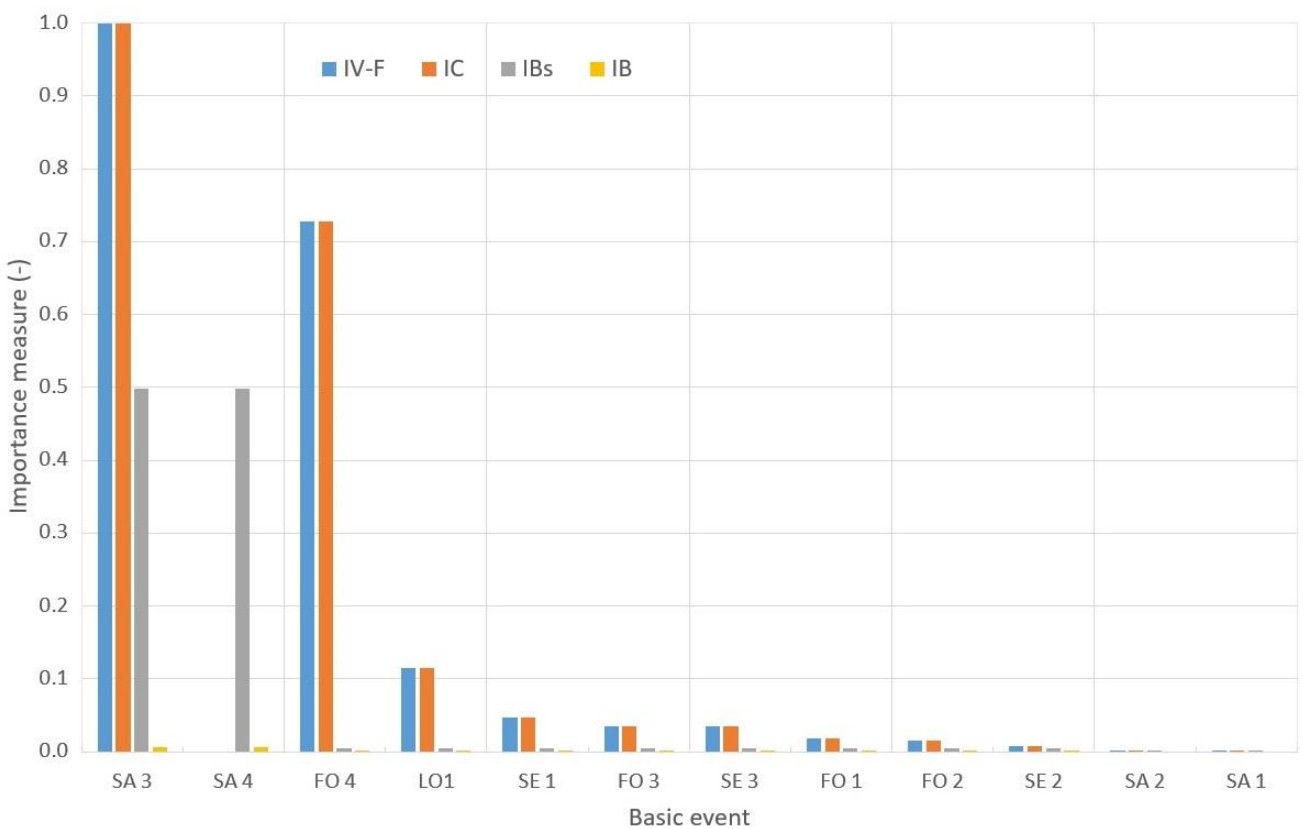

**Figure 8.** Analysis of the importance of the individual basic events.

## 4. Conclusions

Despite the expansion of human knowledge regarding the causes of fires and explosions, the use of appropriate construction materials, the introduction of improved operating procedures, the development of machine diagnostic systems, and the use of devices to minimize the consequences of accidents, the incidents of explosions in starting air manifolds may still occur. With this in mind, it is advisable to perform a continuous expansion of awareness and knowledge of ship mechanism operators, including the use of periodic training and courses, as well as information communicated through popularization and scientific publications.

According to the authors' intention, this article is intended to contribute to the dissemination of good practices in the operation of marine engines. The results obtained from the analysis of the proposed model can be used to map the root causes of explosions in the starting air manifolds relatively well, which points to the main cause of explosions being the improper operation of the starting valves.

Proper and scheduled engine maintenance, and the ongoing inspection of the engine's condition, make it possible to eliminate the risk of an explosion in the starting systems. The following are of primary importance: timely inspections of the engine's starting valves, checking combustion chambers for leaks (to detect, among other things, leaks in the starting

valves), and direct (laboratory analysis) and indirect (ongoing engine diagnosis) control of the quality of fuel used in the power marine engines. The authors anticipate that the proposed model can be further developed by increasing the decomposition level of the system and by preparing detailed models dedicated to the trunk piston engines, inline and forked engines, and engines with indirect and direct control of the starting valves on the cylinders during the engine's starting sequence.

## 5. Patents

- Chybowski, L., Grzebieniak, R.: Method and system for diagnosing the condition of starting air valves of a Diesel engine, preferably the ship's engine. Patent: PL 217302 B1-Polish Patent Office, 19 March 2009.
- Chybowski, L., Kazienko, D., Housing of the detector of diagnostic signals values deviations. Registered design nr 25673 (application Wp.27118, 20 November 2018)—Polish Patent Office, 7 November 2019.
- Chybowski, L., Kazienko, D., Electronic device for detecting deviations of the signal from sensors monitoring the state of the diagnosed object and a way of this detection. European Patent Office Application EP 19460052.4 (zgłoszenie oryginalne P.427649)—Polish Patent Office, 8 November 2019.

**Author Contributions:** Conceptualization, L.C., D.W. and A.J.; methodology, L.C.; software, L.C.; validation, L.C., D.W. and A.J.; formal analysis, L.C.; investigation, L.C., D.W. and A.J.; resources, L.C.; data curation, L.C.; writing—original draft preparation, L.C., D.W. and A.J.; writing—review and editing, L.C., D.W. and A.J.; visualization, L.C., D.W. and A.J.; supervision, L.C.; project administration, L.C.; funding acquisition, L.C. All authors have read and agreed to the published version of the manuscript.

**Funding:** This research was funded by the Ministry of Science and Higher Education (MEiN) of Poland, grant number 1/S/KPBMiM/22. The APC was funded by MDPI.

**Institutional Review Board Statement:** Not applicable.

**Informed Consent Statement:** Not applicable.

**Data Availability Statement:** All data are presented in the paper.

**Conflicts of Interest:** The authors declare no conflict of interest. The funders had no role in the design of the study; in the collection, analyses, or interpretation of data; in the writing of the manuscript; or in the decision to publish the results.

## Abbreviations

| | |
|---|---|
| And 1 | logical conjunction gate |
| ClassNK | Japanese classification society, Nippon Kaiji Kyokai |
| DNV | Norwegian classification society, Det Norske Veritas |
| ERAC | Exact Reliability/Availability Calculation algorithm |
| FO 1, FO 2, FO 3, FO 4 | basic events associated with the failures in the fuel oil system |
| FTA | Fault Tree Analysis |
| HF 1, HF 2 | house events |
| IACS | International Association of Classification Societies |
| LO 1 | basic event associated with the failure in the lubricating oil system |
| LS | Lloyd's Register of Shipping |
| Or 1, Or 2, Or 3, Or 4, Or 5, Or 6, Or 7, Or 8 | logical disjunction gates |
| RMMV | Royal Mail Motor Vessel |
| SA 1, SA 2, SA 3, SA 4 | basic events associated with the failures in the starting air system |
| SE 1, SE 2, SE 3 | basic events associated with the failures in the scavenging air system |
| SOLAS | International Convention for the Safety of Life at Sea |
| T 1 | transfer (interconnection) between the subtree and the main fault tree |
| T/C | turbocharger |

| | |
|---|---|
| VLCC | Very Large Crude Carrier |
| Symbols: | |
| $A$ | set of system states |
| $A_0(t)$ | system availability |
| $f_0$ | frequency of the top event |
| $i$ | event enumerator |
| $I^B(t)$ | Birnbaum measure of reliability importance |
| $I^{Bs}(t)$ | Birnbaum measure of structural importance |
| $I^C(t)$ | Criticality (Lambert's) measure of importance |
| $I^{V-F}(t)$ | Vesely–Fussell's measure of importance |
| $j$ | cut set enumerator |
| $p_i, q_i$ | auxiliary variables |
| $\breve{Q}_j(t)$ | the probability of J-th cut set occurrence |
| $Q_0(t)$ | system unavailability |
| $R_0(t)$ | system reliability |
| $t$ | mission time |
| T | mean time to system failure |
| $T_r$ | mean time to repair |
| $\lambda$ | failure intensity |
| $\vec{y}$ | random state vector of input events |
| $y_i$ | state of $i$-th event |

## Appendix A

**Table A1.** Summary of the selected starting air manifold explosions.

| Date of Event | Information about the Ship | Engine | Causes of the Explosion | Effects of Explosion and Source |
|---|---|---|---|---|
| 17 October 1960 | RMMV Capetown Castle passenger liner | Left main engine Harland and Wolff B&W 10. 660/1500 D.A. (10-cylinder, two-stroke, slow-speed, reversible crosshead engine) | Leaky starting valves. Hydraulic shock of the oil in the start-up system, which is a residue after the attempt to unblock the vent lines of the start-up system was carried out using oil supplied under pressure with a pump | Death of seven machine crew members [13] |
| Before 31 August 1973 | Transport ship at the entrance to Kobe port | One of two six-cylinder, four-stroke, trunk piston engines with a power of 2471 kW each | Defective starting valves | Damage to the engine's starting system components. Loss of ship propulsion [12] |
| 6 February 1978 | Bulk carrier m/v En Gedi | Main engine SA Fiat SGM-Torino | Not determined | Bursting of starting air lines on cylinder systems 2, 6, 7, and 8 [14] |
| Before 30 June 1979 | Transport ship at the entrance of Shanghai port | Main engine, six-cylinder, two-stroke internal combustion engine with an output of 2794 kW | Defective starting valves | Damage to the engine's starting system components [12] |
| 23 May 1980 | m/t Riva I Tanker | Main engine Eriksbergs Mekaniska Verkstads AB-Sweden | Not determined | Damaged two starting air valves on cylinders, associated piping, air line between cylinders 1 and 2, and perforation of the oil tank surrounding the engine [14] |
| 26 July 1982 | Sweet Grace passenger and cargo ship | Auxiliary engine of generator set No. 2, nominal power 160 kW | Explosion of the starting air distributor | Death of the third mechanic and injury to four other crew members [14] |

**Table A1.** *Cont.*

| Date of Event | Information about the Ship | Engine | Causes of the Explosion | Effects of Explosion and Source |
|---|---|---|---|---|
| Before 30 April 2005 | Ship of the offshore fleet | Auxiliary engine of the generator set | Leaky starting valves. Oil from the air compressor got into the starting air system. The engine was not equipped with a safety valve and flame arrester in the starting compressed air supply system | Mechanical damage to starboard engine room starting system components [15] |
| Before 30 June 2009 | Product vessel | Main engine, two-stroke, slow-speed crosshead engine | Leaky starting valves. Oil from the air compressor got into the starting air system. Defective starting air distributor | Damage to the engine's starting system components [16] |
| 10 August 2013 | Transport vessel | Main engine, a two-stroke, slow-speed crosshead engine made by MAN B&W | Defective starting valve | Damage to the engine's starting system components [17] |
| Before 13 August 2013 | Transport vessel | Main engine, eight-cylinder, two-stroke, slow-speed, crosshead engine | Leaky starting valves. Corrosion inside the starting air supply line to the engine | Loss of propulsion and ship running aground ([18] based on DNV data) |
| Before 31 December 2018 | Transport vessel | Main engine, two-stroke, slow-speed, MC/MC-C or ME/ME-C-type, MAN B&W crosshead engine | Leaky starting valves. Oil from the air compressor got into the starting air system. Improper starting valve seals. Obstructed drainage pipes | Damage to the engine's starting system components [19] |

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
