# Peer review of "The Impact of Marine Engine Component Failures upon an Explosion in the Starting Air Manifold"

_jmse, doi:10.3390/jmse10121850_

Round 1
Reviewer 1 Report
The authors present a research on “The impact of marine engine component failures upon an explosion in the starting air manifold” is interesting work and study begin from the frequency of explosions in the marine engine’s starting air manifolds is determined under real conditions. A cause-and-effect analysis of these explosions and their root causes are identified. This article is intended to contribute to the dissemination of good practices in the operation of marine engines. The results obtained from the analysis of the proposed model can be used to map the root causes of explosions in the starting air manifolds relatively well, which points to the main cause of explosions being the improper operation of the starting valves.
Following observation for improvement:
1) The overall paper representation seems it is not a research article, it is simply a collection of data through rich experience authors have in related field. It is not clear what research objectives is? And what methodology they have used to target the research work.
2) In the introduction section, different conditions are not clearly stated and it looks like a simple collection of data with no analysis. Objective, methodology and organisation of the paper is missing in the introduction section.
3) Figures 1.1, figure 1 is mentioned in the document and it started with figure 2, please correct.
4) In the introduction section, different conditions are not clearly stated and it looks like a simple collection of data.
5) Quality of figures 4 and 6 is poor need to improve for better illustrations.
6) Detailed justification with approach is expected for the probability of an explosion in the starting air manifold during the first year of engine operation (to the first failure).
7) All the tables have multiple pieces of information and from a reader's perspective, it is very difficult to understand so elaborate the work. The tabular data must be compared with the similar work and prove your novelty.
8) The proposed methodology is too narrow and may be elaborated with comparison.
9) Validation of data is required either with similar research work or different model.
10) The author may refer following works with elaborated methodology and analysis for automotive system design and system integration.
a. https://doi.org/10.1016/j.jclepro.2022.132430
b. https://doi.org/10.1007/s11277-020-07853-7
11) The conclusion is delivered with limited content. For this manuscript, the conclusion should be provided with more details, and the future scope are not motioned in this section.
12) The novelty of the work must be addressed and discussed, compare your research with existing research findings and highlight novelty, (compare your work with existing research findings and highlight novelty).
13) Add some quantitative research outcomes in the abstract section.
14) What is the need to mention patent details in the manuscript?
Author Response
Dear Reviewer #1,
We are very grateful for all your comments. Your suggestions have been incorporated into the text, and our detailed responses can be found below. The changes in the paper are marked in red. We hope that you will find the revised version of this manuscript suitable for publication and look forward to hearing from you soon.
Best regards
Authors
Authors’ Responses:
The authors present a research on “The impact of marine engine component failures upon an explosion in the starting air manifold” is interesting work and study begin from the frequency of explosions in the marine engine’s starting air manifolds is determined under real conditions. A cause-and-effect analysis of these explosions and their root causes are identified. This article is intended to contribute to the dissemination of good practices in the operation of marine engines. The results obtained from the analysis of the proposed model can be used to map the root causes of explosions in the starting air manifolds relatively well, which points to the main cause of explosions being the improper operation of the starting valves.
Thank you very much for the positive feedback. At the same time we thank you for all your comments and suggestions, which make it possible to improve the quality of the manuscript. The article has been modified according to the reviewer's suggestions.
Following observation for improvement:
1) The overall paper representation seems it is not a research article, it is simply a collection of data through rich experience authors have in related field. It is not clear what research objectives is? And what methodology they have used to target the research work.
We have added a justification for research to the Introduction. We have also changed the titles and numeration of the sections to improve the quality of communication.
2) In the introduction section, different conditions are not clearly stated and it looks like a simple collection of data with no analysis. Objective, methodology and organisation of the paper is missing in the introduction section.
We have included sections 2-4 to the Introduction. Currently the paper is in line with the IMRAD standard.
3) Figures 1.1, figure 1 is mentioned in the document and it started with figure 2, please correct.
Thank you, we have added this figure.
4) In the introduction section, different conditions are not clearly stated and it looks like a simple collection of data.
We have included sections 2-4 to the Introduction. Currently the paper is in line with the IMRAD standard.
5) Quality of figures 4 and 6 is poor need to improve for better illustrations.
Thank you, we have changed and zoomed the figures.
6) Detailed justification with approach is expected for the probability of an explosion in the starting air manifold during the first year of engine operation (to the first failure).
We have added an explanation to the renamed section 1.1. Background and aims of the research.
7) All the tables have multiple pieces of information and from a reader's perspective, it is very difficult to understand so elaborate the work. The tabular data must be compared with the similar work and prove your novelty.
Table 1 presents the minimum amount of data required for appropriate modeling of the events including markings, types, parameters, values and description. The data source has been provided in the caption. A comparative analysis between different sources has been done in dozens of papers and it is not the authors’ intention to elaborate on it as it is not the main topic of the paper. Tables 2 and 3 have been modified and updated by the authors.
8) The proposed methodology is too narrow and may be elaborated with comparison.
We hope the provided explanation of the methods we used have improved the quality of the paper.
9) Validation of data is required either with similar research work or different model.
To our best knowledge the paper is one of the first to ever attempt to build a probabilistic model of explosion in the marine engine starting air manifold (line). We do not know any similar works which we could compare our research with. Nevertheless we calculated and provided standard errors for estimated indicators, which will hopefully improve the credibility of the presented data.
10) The author may refer following works with elaborated methodology and analysis for automotive system design and system integration.
- https://doi.org/10.1016/j.jclepro.2022.132430
- https://doi.org/10.1007/s11277-020-07853-7
Thank you, we have added the suggested references to the Materials and methods section.
11) The conclusion is delivered with limited content. For this manuscript, the conclusion should be provided with more details, and the future scope are not motioned in this section.
We have provided general remarks in the Conclusion section. We have also pointed out possibilities for future works: “The authors anticipate that the proposed model can be further developed by increasing the decomposition level of the system and by preparing detailed models dedicated to the trunk piston engines, inline and forked engines, and engines with indirect and direct control of the starting valves on the cylinders during the engine’s starting sequence.”
12) The novelty of the work must be addressed and discussed, compare your research with existing research findings and highlight novelty, (compare your work with existing research findings and highlight novelty).
To our best knowledge the paper is one of the first to ever attempt to build a probabilistic model of explosion in the marine engine starting air manifold (line). We do not know any similar works which we could compare our research with.
13) Add some quantitative research outcomes in the abstract section.
Thank you, we have added some quantitative results to the Abstract.
14) What is the need to mention patent details in the manuscript?
According to the information in the publisher’s template, this section is not mandatory but may be added if there are patents resulting from the work reported in this manuscript. The mentioned patents are directly connected with the improvement of engine safety in terms of starting air valves condition monitoring. We have added an appropriate explanation in the Introduction.

Reviewer 2 Report
The authors presented their research work with great effort which is interesting to the readers. The manuscript is recommended for publication in the Journal of Marine Science and Engineering. However, the following points may be incorporated in the revised manuscript before publication.
1. Remove the key words which are not significant and use the most relevant one.
Eg. starting airline; starting air manifold; and root causes; cause and effect analysis; fault tree analysis are repeated in different forms.
2. Present the Fig. 5, 6, 7 with high resolution instead of an image file.
3. Expand all the abbreviations used in the manuscript and present the nomenclature for better understanding.
4. What are the errors associated with the analysis? Quantify the same.
Author Response
Dear Reviewer #2,
We are very grateful for all your comments. Your suggestions have been incorporated into the text, and our detailed responses can be found below. The changes in the paper are marked in red. We hope that you will find the revised version of this manuscript suitable for publication and look forward to hearing from you soon.
Best regards
Authors
Authors’ Responses:
The authors presented their research work with great effort which is interesting to the readers. The manuscript is recommended for publication in the Journal of Marine Science and Engineering. However, the following points may be incorporated in the revised manuscript before publication.
Thank you very much for the positive reception of our article. At the same time we thank you for all your comments and suggestions, which make it possible to improve the quality of the manuscript. The article has been modified according to the reviewer's suggestions.
- Remove the key words which are not significant and use the most relevant one.
Eg. starting airline; starting air manifold; and root causes; cause and effect analysis; fault tree analysis are repeated in different forms.
Thank you, we have corrected the list of keywords.
- Present the Fig. 5, 6, 7 with high resolution instead of an image file.
Thank you, we have changed and zoomed the figures.
- Expand all the abbreviations used in the manuscript and present the nomenclature for better understanding.
Thank you, we have provided descriptions for the abbreviations and added the nomenclature.
- What are the errors associated with the analysis? Quantify the same.
Thank you, we have provided estimated standard errors in Table 3.

Reviewer 3 Report
The authors used FTA method to build a probabilistic model of an explosion in the starting air manifold of a marine engine, and used Monte-Carlo simulation method and an ERAC algorithm to calculate the reliability measures. Meanwhile, they analyzed the impact on the explosion incident of each basic event by using four indicators. The study has sufficient arguments, clear data sources, accurate application of analytical methods and correct results, which can provide readers with good reference. However, there are also some omissions:
1. On line 42, the author points out " these are indicated in Figure 1.1 ". But there is no Figure 1.1.
2. The basis or parameter acquisition process of Table 2 should be supplemented.
3. The sentence in lines 21-22 is not clear, that is, “While the following measures of validity are used …”.
To sum up, the article can be acceptable after minor adjustments.
Author Response
Dear Reviewer #3,
We are very grateful for all your comments. Your suggestions have been incorporated into the text, and our detailed responses can be found below. The changes in the paper are marked in red. We hope that you will find the revised version of this manuscript suitable for publication and look forward to hearing from you soon.
Best regards
Authors
Authors’ Responses:
The authors used FTA method to build a probabilistic model of an explosion in the starting air manifold of a marine engine, and used Monte-Carlo simulation method and an ERAC algorithm to calculate the reliability measures. Meanwhile, they analyzed the impact on the explosion incident of each basic event by using four indicators. The study has sufficient arguments, clear data sources, accurate application of analytical methods and correct results, which can provide readers with good reference. However, there are also some omissions:
Thank you very much for the positive reception of our article. The article has been modified according to the reviewer's suggestions.
- On line 42, the author points out " these are indicated in Figure 1.1 ". But there is no Figure 1.1.
Thank you, we have added this figure.
- The basis or parameter acquisition process of Table 2 should be supplemented.
Thank you for the comment, we have added appropriate information.
- The sentence in lines 21-22 is not clear, that is, “While the following measures of validity are used …”.
Thank you for the comment, we have corrected the sentence.
To sum up, the article can be acceptable after minor adjustments.
Thank you for all your comments and suggestions, which make it possible to improve the quality of the manuscript.
